# Estimating Gaussian Copulas with Missing Data with and without Expert Knowledge

**DOI:** 10.3390/e24121849

**Published:** 2022-12-19

**Authors:** Maximilian Kertel, Markus Pauly

**Affiliations:** 1BMW Group, Battery Cell Competence Centre, 80788 Munich, Germany; 2Department of Statistics, TU Dortmund University, 44227 Dortmund, Germany; 3Research Center Trustworthy Data Science and Security, UA Ruhr, 44227 Dortmund, Germany

**Keywords:** missing at random, expert knowledge, expectation maximization, semiparametric estimation

## Abstract

In this work, we present a rigorous application of the Expectation Maximization algorithm to determine the marginal distributions and the dependence structure in a Gaussian copula model with missing data. We further show how to circumvent a priori assumptions on the marginals with semiparametric modeling. Further, we outline how expert knowledge on the marginals and the dependency structure can be included. A simulation study shows that the distribution learned through this algorithm is closer to the true distribution than that obtained with existing methods and that the incorporation of domain knowledge provides benefits.

## 1. Introduction

Even though the amount of data is increasing due to new technologies, big data are by no means good data. For example, missing values are ubiquitous in various fields, from the social sciences [1] to manufacturing [2]. For explanatory analysis or decision making, one is often interested in the joint distribution of a multivariate dataset, and its estimation is a central topic in statistics [3]. At the same time, there exists background knowledge in many domains that can help to compensate for the potential shortcomings of datasets. For instance, domain experts have an understanding of the causal relationships in the data generation process [4]. It is the scope of this paper to unify expert knowledge and datasets with missing data to derive approximations of the underlying joint distribution.

To estimate the multivariate distribution, we use copulas, where the dependence structure is assumed to belong to a parametric family, while the marginals are estimated nonparametrically. Genest et al. [5] showed that for complete datasets, a two-step approach consisting of the estimation of the marginals with an empirical cumulative distribution function (ecdf) and subsequent derivation of the dependence structure is consistent. This idea is even transferable to high dimensions [6].

In the case of missing values, the situation becomes more complex. Here, nonparametric methods do not scale well with the number of dimensions [7]. On the other hand, assuming that the distribution belongs to a parametric family, it can often be derived by using the EM algorithm [8]. However, this assumption is, in general, restrictive. Due to the encouraging results for complete datasets, there have been several works that have investigated the estimation of the joint distribution under a copula model. The authors of [9,10] even discussed the estimation in a missing-not-at-random (MNAR) setting. While MNAR is less restrictive than missing at random (MAR), it demands the explicit modeling of the missing mechanism [11]. On the contrary, the authors of [12,13] provided results in cases in which data were missing completely at random (MCAR). This strong assumption is rarely fulfilled in practice. Therefore, we assume an MAR mechanism in what follows [11].

Another interesting contribution [14] assumed external covariates, such that the probability of a missing value depended exclusively on them and not on the variables under investigation. They applied inverse probability weighting (IPW) and the two-step approach of [5]. While they proved a consistent result, it is unclear how this approach can be adapted to a setting without those covariates. IPW for general missing patterns is computationally demanding, and no software exists [15,16]. Thus, IPW is mostly applied with monotone missing patterns that appear, for example, in longitudinal studies [17]. The popular work of [18] proposed an EM algorithm in order to derive the joint distribution in a Gaussian copula model with data MAR [11]. However, their approach had weaknesses:The presented algorithm was inexact. Among other things, the algorithm simplified by assuming that the marginals and the copula could be estimated separately (compare Equation (Equation 6) in [18] and Equation (Equation 11) in this paper).If there was no a priori knowledge of the parametric family of all marginals, Ref. [18] proposed using the ecdf of the observed data points. Afterwards, they exclusively derived the parameters of the copula. This estimator of the marginals was biased [19,20], which is often overlooked in the copula literature, e.g., [21] (Section 4.3), [22] (Section 3), [23] (Section 3), or [24] (Section 3).The description of the simulation study was incomplete and the results were not reproducible.

The aim of this paper is to close these gaps, and our contributions are the following:We give a rigorous derivation of the EM algorithm under a Gaussian copula model. Similarly to [5], it consists of two separate steps, which estimate the marginals and the copula, respectively. However, these two steps alternate.We show how prior knowledge about the marginals and the dependency structure can be utilized in order to achieve better results.We propose a flexible parametrization of the marginals when a priori knowledge is absent. This allows us to learn the underlying marginal distributions; see Figure 1.We provide a Python library that implements the proposed algorithm.

The structure of this paper is as follows. In Section 2, we review some background information about the Gaussian copula. We proceed by presenting the method (Section 3). In Section 4, we investigate its performance and the effect of domain knowledge in simulation studies. We conclude in Section 5. All technical aspects and proofs in this paper are given in Appendix A and Appendix B.

## 2. The Gaussian Copula Model

### 2.1. Notation and Assumptions

In the following, we consider a *p*-dimensional dataset $\left\{x^{1},\ldots ,x^{N}\right\}\subset R^{p}$ of size *N*, where $x^{1},\ldots ,x^{N}$ are i.i.d. samples from a *p*-dimensional random vector $X=\left(X_{1},\ldots ,X_{p}\right)$ with a joint distribution function *F* and marginal distribution functions $F_{1},\ldots ,F_{p}$. We denote the entries of $x^{\ell }$ by $x^{\ell }=\left(x_{1}^{\ell },\ldots ,x_{p}^{\ell }\right)\forall \ell =1,\ldots ,N$. The parameters of the marginals are represented by $\theta =\left(\theta_{1},\ldots ,\theta_{p}\right)$, where $\theta_{j}$ is the parameter of $F_{j}$, so we write $F_{j}^{\theta_{j}}$, where $\theta_{j}$ can be a vector itself.

For ℓ∈{1,…,p}, we define obs(ℓ)⊂{1,…,p} as the index set of the observed and mis(ℓ)⊂{1,…,p} as the index set of the missing columns of $x^{\ell }$. Hence, mis(ℓ)∪obs(ℓ)={1,…,p} and mis(ℓ)∩obs(ℓ)=∅. $R=\left(R_{1},\ldots ,R_{p}\right)\in {\left\{0,1\right\}}^{p}$ is a random vector for which $R_{i}=0$ if $X_{i}$ is missing and $R_{i}=1$ if $X_{i}$ can be observed. Further, we define ϕ to be the density function and Φ to be the distribution function of the one-dimensional standard normal distribution. $\Phi_{\mu ,\Sigma }$ stands for the distribution function of a *p*-variate normal distribution with covariance $\Sigma \in R^{p\times p}$ and mean $\mu \in R^{p}$. To simplify the notation, we define $\Phi_{\Sigma }:=\Phi_{0,\Sigma }$. For a matrix $A\in R^{p\times p}$, the entry of the *i*-th row and the *j*-th column is denoted by $A_{ij}$, while for index sets S,T⊂{1,…,p}, $A_{S,T}$ is the submatrix of *A* with the row number in S and column number in T. For a (random) vector x (X), $x_{S}$ ($X_{S}$) is the subvector containing entries with the index in S.

Throughout, we assume *F* to be strictly increasing and continuous in every component. Therefore, $F_{j}$ is strictly increasing and continuous for all j∈{1,…,p}, and so is the existing inverse function $F_{j}^{-1}$. For $S=\left\{s_{1},\ldots ,s_{k}\right\}\subset \left\{1,\ldots ,p\right\}$, we define $F_{S}:R^{\left|S\right|}\to R^{\left|S\right|}$ by
$$F_{S}\left(x_{s_{1}},\ldots ,x_{s_{k}}\right)=\left(F_{s_{1}}\left(x_{s_{1}}\right),\ldots ,F_{s_{k}}\left(x_{s_{k}}\right)\right).$$

This work assumes that data are Missing at Random (MAR), as defined by [11], i.e.,
(1)$$P_{X,R}\left(R=r|X_{r}=x_{-r},X_{r}=x_{r}\right)=P_{X,R}\left(R=r|X_{r}=x_{r}\right),$$
where $X_{r}:=X_{\left\{i:\;r_{i}=1\right\}}$ are the observed and $X_{-r}:=X_{\left\{i:\;r_{i}=0\right\}}$ are the missing entries of X.

### 2.2. Properties

Sklar’s theorem [25] decomposes *F* into its marginals $F_{1},\ldots ,F_{p}$ and its dependency structure *C* with
(2)$$F\left(x_{1},\ldots ,x_{p}\right)=C\left(F_{1}\left(x_{1}\right),\ldots ,F_{p}\left(x_{p}\right)\right).$$

Here, *C* is a copula, which means it is a *p*-dimensional distribution function with support ${\left[0,1\right]}^{p}$ whose marginal distributions are uniform. In this paper, we focus on Gaussian copulas, where
(3)$$C_{\Sigma }\left(u_{1},\ldots ,u_{p}\right)=\Phi_{\Sigma }\left(\Phi^{-1}\left(u_{1}\right),\ldots ,\Phi^{-1}\left(u_{p}\right)\right)$$
and Σ is a covariance matrix with $\Sigma_{jj}=1\;\forall j\in \left\{1,\ldots ,p\right\}$. Beyond all multivariate normal distributions, there are distributions with non-normal marginals whose copula is Gaussian. Hence, the Gaussian copula model provides an extension of the normality assumption. Consider a random vector X whose copula is $C_{\Sigma }$. Under the transformation
$$Z:=\Phi^{-1}\circ F\left(X\right):=\left(\Phi^{-1}\circ F_{1}\left(X_{1}\right),\ldots ,\Phi^{-1}\circ F_{p}\left(X_{p}\right)\right),$$
it holds that
(4)$$\begin{array}{cc} F_{Z}\left(z_{1}\ldots ,z_{p}\right)\; & =P\left(Z_{1}\leq z_{1},\ldots ,Z_{p}\leq z_{p}\right)\\ \; & =P\left(X_{1}\leq F_{1}^{-1}\left(\Phi \left(z_{1}\right)\right),\ldots ,X_{p}\leq F_{p}^{-1}\left(\Phi \left(z_{p}\right)\right)\right)\\ \; & =\Phi_{\Sigma }\left(\Phi^{-1}\left(F_{1}\left(F_{1}^{-1}\left(\Phi (z_{1})\right)\right)\right),\ldots ,\Phi^{-1}\left(F_{p}\left(F_{p}^{-1}\left(\Phi (z_{p})\right)\right)\right)\right)\\ \; & =\Phi_{\Sigma }\left(z_{1},\ldots ,z_{p}\right) \end{array}$$
and hence, Z is normally distributed with mean 0 and covariance Σ. The two-step approaches given in [5,6] use this property and apply the following scheme:Find consistent estimates $\hat{F_{1}},\ldots ,\hat{F_{p}}$ for the marginal distributions $F_{1},\ldots ,F_{p}$.Find Σ by estimating the covariance of the random vector
$$Z=\left(\Phi^{-1}\left(\hat{F_{1}}\left(X_{1}\right)\right),\ldots ,\Phi^{-1}\left(\hat{F_{p}}\left(X_{p}\right)\right)\right).$$

From now on, we assume that the marginals of X have existing density functions $f_{1},\ldots ,f_{p}$. Then, by using Equation (Equation 4) and a change of variables, we can derive the joint density function
(5)$$f_{F_{1},\ldots ,F_{p},\Sigma }\left(x_{1},\ldots ,x_{p}\right)=f\left(x_{1},\ldots ,x_{p}\right)={\left|\Sigma \right|}^{-\frac{1}{2}}\exp\left(-\frac{1}{2}z^{T}\left(\Sigma^{-1}-I\right)z\right)\prod_{j=1}^{p}f_{j}\left(x_{j}\right),$$
where $z:=\left(\Phi^{-1}\left(F_{1}\left(x_{1}\right)\right),\ldots ,\Phi^{-1}\left(F_{p}\left(x_{p}\right)\right)\right)$. As for the multivariate normal distribution, we can identify the conditional independencies ([6]) from the inverse of the covariance matrix $K:=\Sigma^{-1}$ by using the property
(6)$$K_{jk}=K_{kj}=0\Leftrightarrow X_{j}⊥X_{k}|\left\{X_{i}:i\in \left\{1,\ldots ,p\right\}\setminus \left\{j,k\right\}\right\}.$$

*K* is called the precision matrix. In order to slim down the notation, we define
$$\Phi^{-1}\left(F_{S}\left(x_{S}\right)\right):=\left(\Phi^{-1}\left(F_{s_{1}}\left(x_{s_{1}}\right)\right),\ldots ,\Phi^{-1}\left(F_{s_{k}}\left(x_{s_{k}}\right)\right)\right)$$
and similarly
$$F_{S}^{-1}\left(\Phi (z_{S})\right):=\left(F_{s_{1}}^{-1}\left(\Phi \left(z_{s_{1}}\right)\right),\ldots ,F_{s_{1}}^{-1}\left(\Phi \left(z_{s_{k}}\right)\right)\right).$$

The former function transforms the data of a Gaussian copula distribution to be normally distributed. The latter mapping takes multivariate normally distributed data and returns data following a Gaussian copula distribution with marginals $F_{s_{1}},\ldots ,F_{s_{k}}$. The conditional density functions have a closed form.

**Proposition** **1**(Conditional Distribution of Gaussian Copula). *Let $S=\{s_{1},\ldots ,s_{k}\}$ and $T=\{t_{1},\ldots ,t_{k^{\prime }}\}$ be such that $T\dot{\cup }S=\left\{1,\ldots ,p\right\}$.*
*The conditional density of $X_{T}|X_{S}=x_{S}$ is given by*$$f\left(x_{T}|X_{S}=x_{S}\right)={\left|\Sigma^{\prime }\right|}^{-\frac{1}{2}}\exp\left(-\frac{1}{2}{\left(z_{T}-\mu \right)}^{T}\Sigma^{\prime -1}\left(z_{T}-\mu \right)\right)\exp\left(\frac{1}{2}z_{T}^{T}z_{T}\right)\prod_{j\in T}f_{j}\left(x_{j}\right),$$*where $\mu =\Sigma_{T,S}\Sigma_{S,S}^{-1}z_{S}$, $\Sigma^{\prime }=\Sigma_{T,T}-\Sigma_{T,S}\Sigma_{S,S}^{-1}\Sigma_{S,T}$, $z_{T}=\Phi^{-1}\left(F_{T}\left(x_{T}\right)\right)$  and $z_{S}=\Phi^{-1}\left(F_{S}\left(x_{S}\right)\right).$**$\Phi^{-1}\left(F_{T}\left(X_{T}\right)\right)|X_{S}=x_{s}$ is normally distributed with mean ***μ*** and covariance $\Sigma^{\prime }$.**The expectation of $h\left(X_{T}\right)$ with respect to the density $f(x_{T}|X_{S}=x_{S})$ can be expressed by*$$\int h\left(x_{T}\right)f\left(x_{T}|X_{S}=x_{S}\right)dx_{T}=\int h\left(F_{T}^{-1}\left(\Phi \left(z_{T}\right)\right)\right)\phi_{\mu ,\Sigma^{\prime }}\left(z_{T}\right)dz_{T}.$$

Proposition 1 shows that the conditional distribution’s copula is Gaussian as well. More importantly, we can derive an algorithm for sampling from the conditional distribution.
**Algorithm 1:**Sampling from the conditional distribution of a Gaussian copula**Input**: $x_{S},\Sigma ,F_{1},\ldots ,F_{p}$**Result**: *m* samples of $X_{T}|X_{S}=x_{S}$Calculate $z_{S}:=\Phi^{-1}\left(F_{S}\left(x_{S}\right)\right)$Calculate μ and $\Sigma^{\prime }$ as in Proposition 1 using $z_{S}$ and ΣDraw samples $\left\{z^{1},\ldots ,z^{m}\right\}$ from $N(\mu ,\Sigma^{\prime })$**return** {$F_{T}^{-1}\left(\Phi (z^{1})\right),\ldots ,F_{T}^{-1}\left(\Phi (z^{m})\right)\}$

The very last step follows with Proposition 1, as it holds for any measurable $A\subset R^{k^{\prime }}$:$$\begin{array}{cc} P\left(X_{T}\in A|X_{S}=x_{S}\right) & =\int 1_{A}\left(x_{T}\right)f\left(x_{T}|X_{S}=x_{S}\right)dx_{T}=\int 1_{A}\left(F_{T}^{-1}\left(\Phi \left(z_{T}\right)\right)\right)\phi_{\mu ,\Sigma^{\prime }}\left(z_{T}\right)dz_{T}. \end{array}$$

## 3. The EM Algorithm in the Gaussian Copula Model

### 3.1. The EM Algorithm

Let $\left\{y^{1},\ldots ,y^{N}\right\}\subset R^{p}$ be a dataset following a distribution with parameter ψ and corresponding density function $g_{\psi }\left(\;\cdot \;\right)$, where observations are MAR. The EM algorithm [8] finds a local optimum of the log-likelihood function
$$\begin{array}{cc} \sum_{\ell =1}^{N}\ln\left(g_{\psi }\left(y_{\mathrm{obs}(\ell )}^{\ell }\right)\right) & =\sum_{\ell =1}^{N}\int \ln\left(g_{\psi }\left(\left(y_{\mathrm{obs}(\ell )}^{\ell },y_{\mathrm{mis}(\ell )}\right)\right)\right)\\ \; & \;\;g_{\psi }\left(y_{\mathrm{mis}(\ell )}|Y_{\mathrm{obs}(\ell )}^{\ell }=y_{\mathrm{obs}(\ell )}^{\ell }\right)dy_{\mathrm{mis}(\ell )}\\ \; & =\sum_{\ell =1}^{N}E_{\psi }\left(\ln\left(g_{\psi }\left(\left(y_{\mathrm{obs}(\ell )},y_{\mathrm{mis}(\ell )}\right)\right)\right)|Y_{\mathrm{obs}}^{\ell }=y_{\mathrm{obs}(\ell )}^{\ell }\right). \end{array}$$

After choosing a start value $\psi^{0}$, it does so by iterating the following two steps.

E-Step: Calculate
(7)$$\begin{array}{cc} \lambda (\psi |y^{1},\ldots ,y^{N},\psi^{t}) & :=\sum_{\ell =1}^{N}E_{\psi^{t}}\left(\ln\left(g_{\psi }\left(\left(y_{\mathrm{obs}(\ell )},y_{\mathrm{mis}(\ell )}\right)\right)\right)|Y_{\mathrm{obs}}^{\ell }=y_{\mathrm{obs}(\ell )}^{\ell }\right)\\ \; & =\sum_{\ell =1}^{N}\lambda \left(\psi |y^{\ell },\psi^{t}\right). \end{array}$$M-Step: Set
(8)$$\psi^{t+1}=\underset{\psi }{\mathrm{argmax}}\lambda \left(\psi |y^{1},\ldots ,y^{N},\psi^{t}\right)$$
and t=t+1.

For our purposes, there are two extensions of interest:If there is no closed formula for the right-hand side of Equation (Equation 7), one can apply Monte Carlo integration [26] as an approximation. This is called the Monte Carlo EM algorithm.If $\psi =\left(\psi_{1},\ldots ,\psi_{v}\right)$ and the joint maximization of (Equation 8) with respect to ψ is not feasible, Ref. [27] proposed a sequential maximization. Thus, we optimize (Equation 8) with respect to $\psi_{i}$ while holding $\psi_{1}=\psi_{1}^{t+1},\ldots ,\psi_{i-1}=\psi_{i-1}^{t+1},\psi_{i+1}=\psi_{i+1}^{t},\ldots ,\psi_{v}=\psi_{v}^{t}$ fixed before we continue with $\psi_{i+1}$. This is called the Expectation Conditional Maximization (ECM) algorithm.

### 3.2. Applying the ECM Algorithm on the Gaussian Copula Model

As we need a full parametrization of the Gaussian copula model for the EM algorithm, we assume parametric marginal distributions $F_{1}^{\theta_{1}},\ldots ,F_{p}^{\theta_{p}}$ with densities $f_{1}^{\theta_{1}},\ldots ,f_{p}^{\theta_{p}}$. According to Equation (Equation 5), the joint density with respect to the parameters $\theta =\left(\theta_{1},\ldots ,\theta_{p}\right)$ and Σ has the form
(9)$$f_{\theta ,\Sigma }\left(x_{1},\ldots ,x_{p}\right)={\left|\Sigma \right|}^{-\frac{1}{2}}\exp\left(-\frac{1}{2}z_{\theta }^{T}\left(\Sigma^{-1}-I\right)z_{\theta }\right)\prod_{j=1}^{p}f_{j}^{\theta_{j}}\left(x_{j}\right),$$
where $z_{\theta }:=\left(\Phi^{-1}\left(F_{1}^{\theta_{1}}\left(x_{1}\right)\right),\ldots ,\Phi^{-1}\left(F_{p}^{\theta_{p}}\left(x_{p}\right)\right)\right)$. Section 3.3 will describe how we can keep the flexibility for the marginals despite the parametrization. However, first, we outline the EM algorithm for general parametric marginal distributions.

#### 3.2.1. E-Step

Set $K:=\Sigma^{-1}$ and $K^{t}:=\Sigma^{t^{-1}}$. For simplicity, we pick one summand in Equation (Equation 7). By Equation (Equation 7) and (Equation 9), it holds with $\psi =\left(\theta ,\Sigma \right)$ and $x^{\ell }$ taking the role of $y^{\ell }$:(10)$$\begin{array}{cc} \lambda (\theta ,\Sigma |x^{\ell },\theta^{t},\Sigma^{t}) & =E_{\theta^{t},\Sigma^{t}}\left(\ln\left(f_{\theta ,\Sigma }\left(\left(x_{\mathrm{obs}(\ell )},x_{\mathrm{mis}(\ell )}\right)\right)\right)|X_{\mathrm{obs}(\ell )}=x_{\mathrm{obs}(\ell )}^{\ell }\right)\\ \; & =-\frac{1}{2}\ln\left(|\Sigma |\right)\\ \; & \;-\frac{1}{2}E_{\Sigma^{t},\theta^{t}}\left({z_{\theta }}^{T}\left(K-I\right)z_{\theta }|X_{\mathrm{obs}(\ell )}=x_{\mathrm{obs}(\ell )}^{\ell }\right)\\ \; & \;+\sum_{j=1}^{p}E_{\Sigma^{t},\theta^{t}}\left(\ln\left(f_{j}^{\theta_{j}}\left(x_{j}\right)\right)|X_{\mathrm{obs}(\ell )}=x_{\mathrm{obs}(\ell )}^{\ell }\right). \end{array}$$

The first and last summand depend only on Σ and θ, respectively. Thus, of special interest is the second summand, for which we obtain the following with Proposition 1:(11)$$E_{\Sigma^{t},\theta^{t}}\left(z_{\theta }^{T}\left(K-I\right)z_{\theta }|X_{\mathrm{obs}(\ell )}=x_{\mathrm{obs}(\ell )}^{\ell }\right)=\int \left(z_{\theta ,\theta^{t}}^{T}\left(K-I\right)z_{\theta ,\theta^{t}}\right)\phi_{\mu ,{\Sigma^{t}}^{\prime }}\left(q_{\mathrm{mis}(\ell )}\right)dq_{\mathrm{mis}(\ell )},$$
where
$$z_{\theta ,\theta^{t}}:=\left(\Phi^{-1}\left(F_{1}^{\theta_{1}}\left(F_{1}^{{\theta_{1}^{t}}^{-1}}\left(\Phi (q_{1})\right)\right)\right),\ldots ,\Phi^{-1}\left(F_{p}^{\theta_{p}}\left(F_{p}^{{\theta_{p}^{t}}^{-1}}\left(\Phi (q_{p})\right)\right)\right)\right).$$

Here,
$$\mu =\Sigma_{\mathrm{mis}(\ell ),\mathrm{obs}(\ell )}\Sigma_{\mathrm{obs}(\ell ),\mathrm{obs}(\ell )}^{-1}\Phi^{-1}\left(F_{\mathrm{obs}(\ell )}^{\theta^{t}}\left(x_{\mathrm{obs}(\ell )}^{\ell }\right)\right)$$
and
$${\Sigma^{t}}^{\prime }=\Sigma_{\mathrm{mis}(\ell ),\mathrm{mis}(\ell )}^{t}-\Sigma_{\mathrm{mis}(\ell ),\mathrm{obs}(\ell )}^{t}{\left(\Sigma_{\mathrm{obs}(\ell ),\mathrm{obs}(\ell )}^{t}\right)}^{-1}\Sigma_{\mathrm{obs}(\ell ),\mathrm{mis}(\ell )}^{t}.$$

At this point, the authors of [18] neglected that, in general,
$$F_{j}^{\theta_{j}^{t}}\neq F_{j}^{\theta_{j}},j=1,\ldots ,p$$
holds, and hence, (Equation 11) depends not only on Σ, but also on θ. This let us reconsider their approach, as we describe below.

#### 3.2.2. M-Step

The joint optimization with respect to θ and Σ is difficult, as there is no closed form for Equation (Equation 10). We circumvent this problem by sequentially optimizing with respect to Σ and θ by applying the ECM algorithm. The maximization routine is the following.

Set $\Sigma^{t+1}={\mathrm{argmax}}_{\Sigma }\sum_{l=1}^{N}\lambda \left(\theta^{t},\Sigma |x^{\ell },\theta^{t},\Sigma^{t}\right)$.Set $\theta^{t+1}={\mathrm{argmax}}_{\theta }\sum_{l=1}^{N}\lambda \left(\theta ,\Sigma^{t+1}|x^{\ell },\theta^{t},\Sigma^{t}\right)$.

This is a two-step approach consisting of estimating the copula first and the marginals second. However, both steps are executed iteratively, which is typical for the EM algorithm.

#### Estimating Σ

As we are maximizing Equation (Equation 10) with respect to Σ with a fixed $\theta =\theta^{t}$, the last summand can be neglected. By a change-of-variables argument, we show the following in Theorem A1:$$E_{\Sigma^{t},\theta^{t}}\left({z_{\theta^{t}}}^{T}\left(K-I\right)z_{\theta^{t}}|X_{\mathrm{obs}(\ell )}=x_{\mathrm{obs}(\ell )}^{\ell }\right)=tr\left(\Sigma^{-1}V_{\ell }\right),$$
where $V_{\ell }$ depends on $\Sigma^{t}$ and $z_{\theta^{t},\mathrm{obs}(\ell )}=\Phi^{-1}\left(F_{\mathrm{obs}(\ell )}^{\theta^{t}}\left(x_{\mathrm{obs}(\ell )}^{\ell }\right)\right)$. Thus, considering all observations, we search for
(12)$$\begin{array}{cc} \Sigma^{t+1} & =\underset{\Sigma ,\Sigma_{\ell \ell }=1\forall \ell =1,\ldots ,p}{\mathrm{argmax}}\frac{1}{N}\sum_{l=1}^{N}\lambda \left(\theta^{t},\Sigma |x^{\ell },\theta^{t},\Sigma^{t}\right)\\ \; & =\underset{\Sigma ,\Sigma_{\ell \ell }=1\forall \ell =1,\ldots ,p}{\mathrm{argmax}}\frac{1}{N}\sum_{\ell =1}^{N}-\frac{1}{2}\ln\left(|\Sigma |\right)-\frac{1}{2}tr\left(\Sigma^{-1}V_{\ell }\right)\\ \; & =\underset{\Sigma ,\Sigma_{\ell \ell }=1\forall \ell =1,\ldots ,p}{\mathrm{argmax}}-\frac{1}{2}\ln\left(|\Sigma |\right)-\frac{1}{2}tr\left(\Sigma^{-1}\frac{1}{N}\sum_{\ell =1}^{N}V_{\ell }\right), \end{array}$$
which only depends on the statistic $S:=\frac{1}{N}\sum_{\ell =1}^{N}V_{\ell }$. Generally, this maximization can be formalized as a convex optimization problem that can be solved by a gradient descent. However, the properties of this estimator are not understood (for example, a scaling of *S* by $a\in R_{>0}$ leads to a different solution; see Section A.3). To overcome this issue, we instead approximate the solution with the correlation matrix
$$\underset{\Sigma ,\Sigma_{\ell \ell }=1\forall \ell =1,\ldots ,p}{\mathrm{argmax}}-\frac{1}{2}\ln\left(|\Sigma |\right)-\frac{1}{2}tr\left(\Sigma^{-1}S\right)\approx PSP,$$
where $P\in R^{p}$ is the diagonal matrix with entries $P_{jj}=\frac{1}{\sqrt{S_{jj}}},\forall j=1,\ldots ,p$. This was also proposed in [28] (Section 2.2).

In cases in which there is expert knowledge on the dependency structure of the underlying distribution, one can adapt Equation (Equation 12) accordingly. We discuss this in more detail in Section 4.4.

#### Estimating θ

We now focus on finding $\theta^{t+1}$, which is the maximizer of
$$\begin{array}{cc} \sum_{\ell =1}^{N}\lambda \left(\theta ,\Sigma^{t+1}|x^{\ell },\theta^{t},\Sigma^{t}\right) & =\sum_{\ell =1}^{N}E_{\theta^{t},\Sigma^{t}}\left(\ln\left(f_{\theta ,\Sigma^{t+1}}\left(x_{\mathrm{obs}(\ell )},x_{\mathrm{mis}(\ell )}\right)\right)|X_{\mathrm{obs}(\ell )}=x_{\mathrm{obs}(\ell )}^{\ell }\right)\\ \; & =\sum_{\ell =1}^{N}\int \ln\left(f_{\theta ,\Sigma^{t+1}}\left(x_{\mathrm{obs}(\ell )}^{\ell },x_{\mathrm{mis}(\ell )}\right)\right)\\ & \;\;f_{\theta^{t},\Sigma^{t}}\left(x_{\mathrm{mis}(\ell )}|X_{\mathrm{obs}(\ell )}=x_{\mathrm{obs}(\ell )}^{\ell }\right)dx_{\mathrm{mis}(\ell )} \end{array}$$
with respect to θ. As there is, in general, no closed formula for the right-hand side, we use Monte Carlo integration. Again, we start by considering a single observation $x^{\ell }$ to simplify terms. Employing Algorithm 1, we receive *M* samples $x_{\mathrm{mis}(\ell ),1}^{\ell },\ldots ,x_{\mathrm{mis}(\ell ),M}^{\ell }$ from the distribution of $X_{\mathrm{mis}(\ell )}|X_{\mathrm{obs}(\ell )}=x_{\mathrm{obs}(\ell )}^{\ell }$ given the parameters $\theta^{t}$ and $\Sigma^{t}$. We set $x_{\mathrm{obs}(\ell ),m}^{\ell }=x_{\mathrm{obs}(\ell )}^{\ell }\;\forall m=1,\ldots ,M$. Then, by Equation (Equation 9),
(13)$$\begin{array}{cc} \lambda \left(\theta ,\Sigma^{t+1}|x^{\ell },\theta^{t},\Sigma^{t}\right)\approx C+\frac{1}{M}\sum_{m=1}^{M} & -\frac{1}{2}{\left(\Phi^{-1}\left(F_{1}^{\theta_{1}}\left(x_{1,m}^{\ell }\right)\right),\ldots ,\Phi^{-1}\left(F_{p}^{\theta_{p}}\left(x_{p,m}^{\ell }\right)\right)\right)}^{T}\\ \; & \;\left(K^{t+1}-I\right)\\ \; & \;\left(\Phi^{-1}\left(F_{1}^{\theta_{1}}\left(x_{1,m}^{\ell }\right)\right),\ldots ,\Phi^{-1}\left(F_{p}^{\theta_{p}}(x_{p,m}^{\ell }\right)\right)\\ \; & +\sum_{j=1}^{p}\ln\left(f_{j}^{\theta_{j}}\left(x_{j,m}^{\ell }\right)\right). \end{array}$$

Hence, considering all observations, we set
(14)$$\begin{array}{cc} \theta^{t+1}=\underset{\theta }{\mathrm{argmax}}\frac{1}{M}\sum_{\ell =1}^{N}\sum_{m=1}^{M} & -\frac{1}{2}{\left(\Phi^{-1}\left(F_{1}^{\theta_{1}}\left(x_{1,m}^{\ell }\right)\right),\ldots ,\Phi^{-1}\left(F_{p}^{\theta_{p}}\left(x_{p,m}^{\ell }\right)\right)\right)}^{T}\\ \; & \;\left(K^{t+1}-I\right)\\ \; & \;\left(\Phi^{-1}\left(F_{1}^{\theta_{1}}\left(x_{1,m}^{\ell }\right)\right),\ldots ,\Phi^{-1}\left(F_{p}^{\theta_{p}}(x_{p,m}^{\ell }\right)\right)\\ \; & +\sum_{j=1}^{p}\ln\left(f_{j}^{\theta_{j}}\left(x_{j,m}^{\ell }\right)\right). \end{array}$$

Note that we only use the Monte Carlo samples to update the parameters of the marginal distributions θ. We would also like to point out some interesting aspects about Equations (Equation 13) and (Equation 14):The summand $\sum_{\ell =1}^{N}\sum_{m=1}^{M}\ln\left(f_{j}^{\theta_{j}}\left(x_{j,m}^{\ell }\right)\right)$ describes how well the marginal distributions fit the (one-dimensional) data.The estimations of the marginals are interdependent. Hence, in order to maximize with respect to $\theta_{j}$, we have to take into account all other components of θ.The first summand adjusts for the dependence structure in the data. If all observations at step t+1 are assumed to be independent, then $K^{t+1}=I$, and this term is 0.More generally, the derivative $\frac{\partial \lambda (\theta ,\Sigma^{t+1}|x^{\ell },\theta^{t},\Sigma^{t})}{\partial \theta_{j}}$ depends on $\theta_{k}$ if and only if $K_{jk}^{t+1}\neq 0$. This means that if $\Sigma^{t+1}$ implies the conditional independence of column *j* and *k* given all other columns (Equation (Equation 6)), the optimal $\theta_{j}$ can be found without considering $\theta_{k}$. This, e.g., is the case if we set entries of the precision matrix to 0. Thus, the incorporation of prior knowledge reduces the complexity of the identification of the marginal distributions.

The intuition behind the derived EM algorithm is simple. Given a dataset with missing values, we estimate the dependency structure. With the identified dependency structure, we can derive likely locations of the missing values. Again, these locations help us to find a better dependency structure. This leads to the proposed cyclic approach. The framework of the EM algorithm guarantees the convergence of this procedure to a local maximum for M→∞ in Equation (Equation 14).

### 3.3. Modelling with Semiparametric Marginals

In the case in which the missing mechanism is MAR, the estimation of the marginal distribution using only complete observations is biased. Even worse, any moment of the distribution can be distorted. Thus, one needs a priori knowledge in order to identify the parametric family of the marginals [19,20]. If their family is known, one can directly apply the algorithm of Section 3.2. If this is not the case, we propose the use of a mixture model parametrization of the form
(15)$$F_{j}^{\theta_{j}}\left(x_{j}\right)=\frac{1}{g}\sum_{k=1}^{g}\Phi \left(\frac{x_{j}-\theta_{jk}}{\sigma_{j}}\right),\theta_{j1}\leq \ldots \leq \theta_{jg},\forall j=1,\ldots ,p,$$
where $\sigma_{j}$ is a hyperparameter and the ordering of the $\theta_{jk}$ ensures the identifiability.

Using mixture models for density estimation is a well-known idea (e.g., [29,30,31]). As the authors of [31] noted, mixture models vary between being parametric and being non-parametric, where flexibility increases with *g*. It is reasonable to choose Gaussian mixture models, as their density functions are dense in the set of all density functions with respect to the $L^{1}$-norm [29] (Section 3.2). This flexibility and the provided parametrization make the mixture models a natural choice.

### 3.4. A Blueprint of the Algorithm

The complete algorithm is summarized in Algorithm 2. For the Monte Carlo EM algorithm, Ref. [26] proposed the stabilization of the parameters with a rather small number of samples *M* and to increase this number substantially in the latter steps of the algorithm. This seems to be reasonable for line 2 of Algorithm 2 as well.

If there is no a priori knowledge about the marginals, we propose that we follow Section 3.3. We choose the initial $\theta^{0}$ such that the cumulative distribution function of the mixture model fits the ecdf of the observed data points. For an empirical analysis of the role of *g*, see Section 4.3.3. For $\sigma_{1},\ldots ,\sigma_{p}$, we use a rule of thumb inspired by [3] and set
$$\sigma_{j}=1.06\frac{\hat{\sigma_{j}}}{g^{1/5}},$$
where $\hat{\sigma_{j}}$ is the standard deviation of the observed data points in the *j*-th component.
**Algorithm 2:**Blueprint for the EM algorithm for the Gaussian copula model
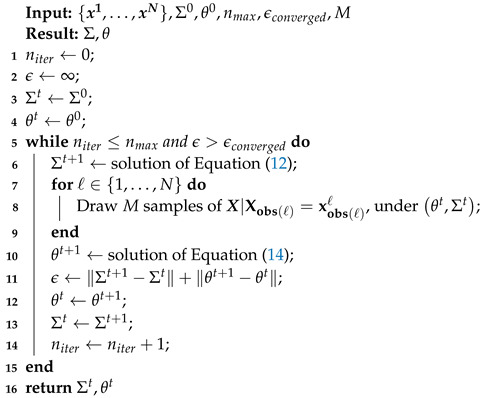


## 4. Simulation Study

We analyze the performance of the proposed estimator in two studies. First, we consider scenarios for two-dimensional datasets and check the potential of the algorithm. In the second part, we explore how expert knowledge can be incorporated and how this affects the behavior and performance. The proposed procedure, which is indexed with EM in the figures below, is compared with:**Standard COPula Estimator (SCOPE)**: The marginal distributions are estimated by the ecdf of the observed data points. This was proposed by [18] if the parametric family is unknown, and it is the state-of-the art approach. Thus, we apply an EM algorithm to determine the correlation structure on the mapped data points
$$z_{j}^{\ell }=\Phi^{-1}\left(\hat{F_{j}}\left(x_{j}^{\ell }\right)\right),\ell =1,\ldots ,N,j=1,\ldots ,p,$$
where $\hat{F_{j}}$ is the ecdf of the observed data points in column *j*. Its corresponding results are indexed with SCOPE in the figures and tables.**Known marginals**: The distribution of the marginals is completely known. The idea is to eliminate the difficulty of finding them. Here, we apply the EM algorithm for the correlation structure on
$$z_{j}^{\ell }=\Phi^{-1}\left(F_{j}\left(x_{j}^{\ell }\right)\right),\ell =1,\ldots ,N,j=1,\ldots ,p,$$
where $F_{j}$ is the real marginal distribution function. Its corresponding results are indexed with a 0 in the figures and tables.**Markov chain–Monte Carlo (MCMC) approach [21]**: The author proposed an MCMC scheme to estimate the copula in a Bayesian fashion. Therefore, Ref. [21] derived the distribution of the multivariate ranks. The marginals are treated as nuisance parameters. We employed the R package sbgcop, which is available on CRAN, as it provides not only a posterior distribution of the correlation matrix Σ, but also imputations for missing values. In order to compare the approach with the likelihood-based methods, we set
$${\hat{\Sigma }}_{MCMC}=\frac{1}{M}\sum_{m=1}^{M}\Sigma^{m},$$
where $\left\{\Sigma^{m}:m=1,\ldots ,M\right\}$ are samples of the posterior distribution of the correlation matrix. For the marginals, we defined
$${\hat{F}}_{j,MCMC}\left(x\right)=\frac{1}{MN}\sum_{\ell =1}^{N}\sum_{m=1}^{M}1_{\left\{x_{j,m}^{\ell }\leq x\right\}},$$
where $x_{j,m}^{\ell }$ is the *m*-th of the total of *M* imputations for $x_{j}^{\ell }$ and $x_{j,m}^{\ell }=x_{j}^{\ell }\;\forall m=1,\ldots ,M$ if $x_{j}^{\ell }$ can be observed. The samples were drawn from the posterior distribution. The corresponding results were indexed with the MCMC approach in the figures and tables.

Sklar’s theorem shows that the joint distribution can be decomposed into the marginals and the copula. Thus, we analyze them separately.

### 4.1. Adapting the EM Algorithm

In Section 4.3 and Section 4.4, we chose g=15, for which we saw a sufficient flexibility. A sensitivity analysis of the procedure with respect to *g* can be found in Section 4.3.3. The initial $\theta^{0}$ was chosen by fitting the marginals to the existing observations, and $\Sigma^{0}$ was the identity matrix. For the number of Monte Carlo samples *M*, we observed that with M=20, θ stabilized after around 10 steps. Cautiously, we ran 20 steps before we increased *M* to 1000, for which we run another five steps. We stopped the algorithm when the condition $∥\Sigma^{t+1}-\Sigma^{t}{∥}_{1}<10^{-5}$ was fulfilled.

### 4.2. Data Generation

We considered a two-dimensional dataset (we would have liked to include the setup of the simulation study of [18]; however, neither could the missing mechanism be extracted from the paper nor did the authors provide it on request) with a priori unknown marginals $F_{1}$ and $F_{2}$, whose copula was Gaussian with the correlation parameter ρ∈[−1,1]. The marginals were chosen to be $\chi^{2}$ with six and seven degrees of freedom. The data matrix $D\in R^{N\times 2}$ kept *N* (complete) observations of the random vector. We enforced the following MAR mechanism:Remove every entry in *D* with probability $0\leq p_{MCAR}<1$. We denote the resulting data matrix (with missing entries) as $D^{MCAR}={\left(D_{\ell j}^{MCAR}\right)}_{\ell =1,\ldots ,N,j=1,2}$.If $D_{\ell 1}^{MCAR}$ and $D_{\ell 2}^{MCAR}$ are observed, remove $D_{\ell 2}^{MCAR}$ with probability
$$\begin{array}{cc} P\left(R_{2}=0|X_{1}=D_{\ell 1},X_{2}=D_{\ell 2}\right) & =P\left(R_{2}=0|X_{1}=D_{\ell 1}\right)\\ \; & =\frac{1}{1+\exp\left(-\beta_{0}-\beta_{1}\Phi^{-1}\left(F_{1}\left(D_{\ell 1}\right)\right)\right)} \end{array}$$We call the resulting data matrix $D^{MAR}$.

The missing patterns were non-monotone. Aside from $p_{MCAR}$, the parameters $\beta_{0}$ and $\beta_{1}$ controlled how many entries were absent in the final dataset. Assuming that ρ>0, $\beta_{1}>0$, and $|\beta_{0}|$ was not too large, the ecdf of the observed values of $X_{2}$ was shifted to the left compared to the true distribution function (changing the signs of $\beta_{1}$ and/or ρ may change the direction of the shift, but the situation is analogous). This can be seen in Figure 1, where we chose N=200, ρ=0.5, $\beta =\left(\beta_{0},\beta_{1}\right)=\left(0,2\right)$. The marginal distribution of $X_{1}$ could be estimated well by the ecdf of the observed data.

### 4.3. Results

This subsection explores how different specifications of the data-generating process presented in Section 4.2 influenced the estimation of the joint distribution. First, we investigate the influence of the share of missing values (controlled via β) and the dependency (controlled via ρ) by fixing the number of observations (denoted by *N*) to 100. Then, we vary *N* to study the behavior of the algorithms for larger sample sizes. Afterwards, we carry out a sensitivity analysis of the EM algorithm with respect to *g*, the number of mixtures. Finally, we study the computational demands of the algorithms.

#### 4.3.1. The Effects of Dependency and Share of Missing Values

We investigate two different choices for the setup in Section 4.2 by setting the parameters to ρ=0.1, β=(−1,1) and ρ=0.5, β=(0,2). For both, we draw 1000 datasets with N=100 each and apply the estimators. To evaluate the methods, we look at two different aspects.

First, we compare the estimators for ρ with respect to bias and standard deviations. The results are depicted in the corresponding third columns of Table 1 and are summarized as boxplots in Figure A1 in Section B.3. We see that no method is clearly superior. While the EM algorithm has a stronger bias for ρ=0.5 than that of SCOPE, it also has a smaller standard deviation. The MCMC approach shows the largest bias. As even known marginals ($\rho_{0}$) do not lead to substantially better estimators compared to SCOPE ($\rho_{SCOPE}$) or the proposed ($\rho_{EM}$) approach, we deduce that (at least in this setting) the estimators for the marginals are almost negligible. MCMC performs notably worse.

Second, we investigate the Cramer–von Mises statistics ω between the estimated and the true marginal distribution ($\omega^{1}$ statistic for the first marginal, $\omega^{2}$ statistic for the second marginal). The results are shown in Table 1 (corresponding first two columns) and are summarized as boxplots in Figure A2 in Section B.3. While for ρ=0.1, the proposed estimator behaves only slightly better than SCOPE, we see that the benefit becomes larger in the case of high correlation and more missing values, especially when estimating the second marginal. This is in line with the intuition that if the correlation is vanishing, the two random variables $X_{1}$ and $X_{2}$ become independent. Thus, $R_{2}$, the missing value indicator, and $X_{2}$ become independent. (Note that there is a difference from the case in which ρ≠0, and hence, the missingness probability $R_{2}$ isconditionally independent from $X_{2}$ given $X_{1}$.) In that case, we can estimate the marginal of $X_{2}$ using the ecdf of the observed data points. Hence, SCOPE’s estimates of the marginals should be good for small values of ρ. An illustration can be found in Figure 2. Again, the MCMC approach performs the worst.

#### 4.3.2. Varying the Sample Size *N*

To investigate the behavior of the methods for larger sample sizes, we repeat the experiment from Section 4.2 with ρ=0.5,β=(0,2) for the sample sizes N=100,200,500,1000. The results are depicted in Table 2 and Figure A3, Figure A4 and Figure A5 in Section B.3. The bias of SCOPE and EM algorithm for ρ seem to vanish for large *N*, while the MCMC approach remains biased. Studying the estimation of the true marginals, the approximation of the second marginal via MCMC and SCOPE improves only slowly and is still poor for the largest sample sizes N=1000. In contrast, the EM algorithm performs best in small sample sizes, and the mean (of $\omega^{1}$ and $\omega^{2}$) and standard deviations (of all three values) move towards 0 for increasing *N*.

#### 4.3.3. The Impacts of Varying the Number of Mixtures *g*

The proposed EM algorithm relies on the hyperparameter *g*, the number of mixtures in Equation (Equation 15). To analyze the behavior of the EM algorithm with respect to *g*, we additionally run the EM algorithm with g=5 and g=30 on the 1000 datasets of Section 4.2 for ρ=0.5, β=(0,2), and N=100. We did not adjust the number of steps in the EM algorithm to keep the results comparable. The results can be found in Table 3. We see that the choice of *g* does not have a large effect on the estimation of ρ. However, an increased *g* leads to better estimates for $X_{1}$. This is in line with the intuition that the ecdf of the first components is an unbiased estimate for the distribution function of $X_{1}$, and setting *g* to the number of samples corresponds to the kernel density estimator. On the other hand, the estimator for $X_{2}$ benefits slightly from g=5, as $\omega_{EM}^{2}$ has a lower mean and standard deviation compared to the choice g=30. However, this effect is small and almost non-existent when we compare g=5 with g=15. As the choice g=15 leads to better estimates of the first marginal compared to g=5, we see this choice as a good compromise for our setting. For applications without prior knowledge, we recommend considering *g* as additional tuning parameter (via cross-validation).

#### 4.3.4. Run Time

We analyze the computational demands of the different algorithms by comparing their run times in the study of Section 4.3.1 with ρ=0.5 and β=(0,2) (the settings ρ=0.1 and β=(−1,1) lead to similar results and are omitted). The run times of all presented algorithms depend not only on the dataset, but also on the parameters (e.g., convergence criterion and $\Sigma^{0}$ for SCOPE). Thus, we do not aim for an extensive study, but focus on the magnitudes. We compare the proposed EM algorithm with a varying number of mixtures (g=5,15,30) with MCMC and SCOPE. The results are shown in Table 4. We see that the EM algorithm has the longest run time, which depends on the number of mixtures *g*. The MCMC approach and the proposed EM algorithm have a higher computational demand than SCOPE, as they are trying to model the interaction between the copula and the marginals. As mentioned in the onset, we could reduce the run time of the EM algorithm by going down to only 10 steps instead of 20.

### 4.4. Inclusion of Expert Knowledge

In the presence of prior knowledge on the dependency structure, the presented EM algorithm is highly flexible. While information on the marginals can be used to parametrize the copula model, expert knowledge on the dependency structure can be incorporated by adapting Equation (Equation 12). In the case of soft constraints on the covariance or precision matrix, one can replace Equation (Equation 12) with a penalized covariance estimation, where the penalty reflects the expert assessment [32,33]. Similarly, one can define a prior distribution on the covariance matrices and set $\Sigma^{t+1}$ as the mode of the posterior distribution (the MAP estimate) of Σ given the statistic *S* of Equation (Equation 12).

Another possibility could be that we are aware of conditional independencies in the data-generating process. This is, for example, the case when causal relationships are known [4]. To exemplify the latter, we consider a three-dimensional dataset X with the Gaussian copula $C_{\Sigma }$ and marginals $X_{1},X_{2},X_{3}$, which are $\chi^{2}$ distributed with six, seven, and five degrees of freedom. The precision is set to
$$K=\Sigma^{-1}=\Delta^{1/2}\left(\begin{array}{ccc} 1 & 0.5 & 0.5\\ 0.5 & 1 & 0\\ 0.5 & 0 & 1 \end{array}\right)\Delta^{1/2},$$
where $\Delta^{1/2}$ is a diagonal matrix, which ensures that the diagonal elements of Σ are 1. We see that $X_{2}$ and $X_{3}$ are conditionally independent given $X_{1}$. The missing mechanism is similar to the one in Section 4.2. The missingness of $X_{3}$ depends on $X_{1}$ and $X_{2}$, while the probability of a missing $X_{1}$ or $X_{2}$ is independent of the others. The mechanism is, again, MAR. Details can be found in Section B.2. We compare the proposed method with prior knowledge on the zeros in the precision matrix (indexed by KP, EM in the figures) with the EM, SCOPE, and MCMC algorithms without background knowledge. We again sample 1000 datasets with 50 observations each from the real distribution. The background knowledge on the precision is used by restricting the non-zero elements in Equation (Equation 12). Therefore, we apply the procedure presented in [34] (Chapter 17.3.1) to find $\Sigma^{t+1}$. The means and standard deviations of the estimates are presented in Table 5.

First, we evaluate the estimated dependency structures by calculating the Frobenius norm of the estimation error $\Sigma -\hat{\Sigma }$. The EM algorithm with background knowledge (KP, EM) performs best and is more stable than its competitors. Apart from MCMC, the other procedures behave similarly, which indicates again that the exact knowledge of the marginal distributions is not too relevant for identifying the dependency structure. MCMC performs the worst.

Second, we see that the proposed EM estimators return marginal distributions that are closer to the truth, while the estimate with background knowledge (KP, EM) performs the best. Thus, the background knowledge on the copula also transfers into better estimates for the marginal distribution—in particular, for $X_{3}$. This is due to Equation (Equation 14) and the comments thereafter. The zeros in the precision structure indicate which other marginals are relevant in order to identify the parameter of a marginal. In our case, $X_{2}$ provides no additional information for $X_{3}$. This information is provided to the EM algorithm through the restriction of the precision matrix.

Finally, we compare the EM estimates of the joint distribution. The relative entropy or Kullback–Leibler divergence is a popular tool for estimating the difference between two distributions [35,36], where one of them is absolutely continuous with respect to the other. A lower number indicates a higher similarity. Due to the discrete structure of the marginals of SCOPE and MCMC, we cannot calculate their relative entropy with respect to the truth. However, we would like to analyze how the estimate of the proposed procedure improves if we include expert knowledge. The results are depicted in Table 6. Again, we observe that the incorporation of extra knowledge improves the estimates. This is in line with Table 5, as the estimation of all components in the joint distribution of Equation (Equation 3) is improved by the domain knowledge.

## 5. Discussion

In this paper, we investigated the estimation of the Gaussian copula and the marginals with an incomplete dataset, for which we derived a rigorous EM algorithm. The procedure iteratively searches for the marginal distributions and the copula. It is, hence, similar to known methods for complete datasets. We saw that if the data are missing at random, a consistent estimate of a marginal distribution depends on the copula and other marginals.

The EM algorithm relies on a complete parametrization of the marginals. The parametric family of the marginals is, in general, a priori unknown and cannot be identified through the observed data points. For this case, we presented a novel idea of employing mixture models. Although this is practically always a misspecification, our simulation study revealed that the combination of our EM algorithm and marginal mixture models delivers better estimates for the joint distribution than currently used procedures do. In principle, uncertainty quantification of the parameters derived by the proposed EM algorithm can be achieved by bootstrapping [37].

There are different possibilities for incorporating expert knowledge. Information on the parametric family of the marginals can be used for their parametrization. However, causal and structural understandings of the data-generating process can also be utilized [4,38,39]. For example, this can be achieved by restricting the correlation matrix or its inverse, the precision matrix. We presented how one can restrict the non-zero elements of the precision, which enforces conditional independencies. Our simulation study showed that this leads not only to an improved estimate for the dependency structure, but also to better estimates for the marginals. This translates into a lower relative entropy between the real distribution and the estimate. We also discussed how soft constraints on the dependency structure can be included.

We note that the focus of this paper is on estimating the joint distribution without precise specification of its subsequent use. Therefore, we did not discuss imputation methods (see, e.g., [40,41,42,43]). However, Gaussian copula models were employed as a device for multiple imputation (MI) with some success [22,24,44]. The resulting complete datasets can be used for inference. All approaches that we are aware of estimate the marginals by using the ecdf of the observed data points. The findings in Section 4 translate into better draws for the missing values.

Additionally, the joint distribution can be utilized for regressing a potentially multivariate Y on Z even if data are missing. By applying the EM algorithm on $X:=\left(Y,Z\right)$ and by Proposition 1, one even obtains the whole conditional distribution of Y given Z=z.

We have shown how to incorporate a causal understanding of the data-generating process. However, in the potential outcome framework of [45], the derivation of a causal relationship can also be interpreted as a missing data problem in which the missing patterns are “misaligned” [46]. Our algorithm is applicable for this.

figuresection tablesection

## Figures and Tables

**Figure 1 entropy-24-01849-f001:**
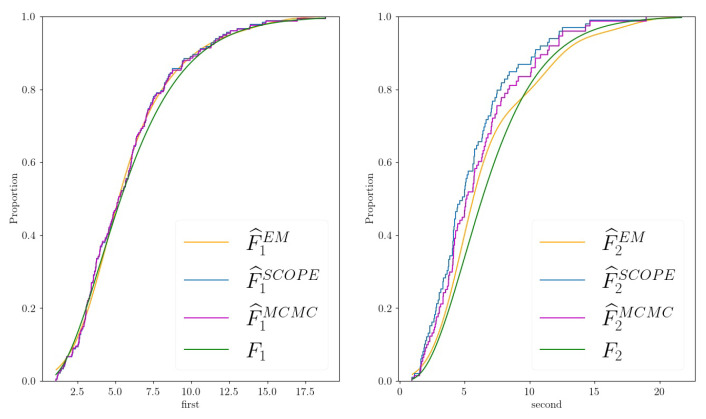
Estimates of the proposed EM algorithm (${\hat{F}}_{i}^{EM}$, orange line), the Standard Copula Estimator (${\hat{F}}_{i}^{SCOPE}$, blue line, corresponds to ecdf), the Markov chain–Monte Carlo approach (${\hat{F}}_{i}^{MCMC}$, purple line) for the marginals $X_{i}$, i=1,2, and the truth ($F_{i}$, green line) of a two-dimensional example dataset generated as described in Section 4.2 with N=200, ρ=0.5, and β=(0,2).

**Figure 2 entropy-24-01849-f002:**
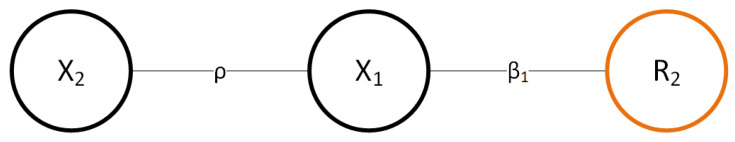
Dependency graph for $X_{1},X_{2}$, and $R_{2}$. $X_{2}$ is independent of $R_{2}$ if either $X_{1}$ and $X_{2}$ are independent (ρ=0) or if $X_{1}$ and $R_{2}$ are independent ($\beta_{1}=0$).

**Table 1 entropy-24-01849-t001:** Comparison of the algorithms with respect to the Cramer–von Mises distance between the estimated and the true first ($\omega^{1}$) and true second marginal distributions ($\omega^{2}$), as well as the correlation (ρ). Shown are the mean and standard deviation of the proposed EM algorithm (EM), the method based on known marginals (0), the Standard Copula Estimator (SCOPE), and the Markov chain–Monte Carlo approach (MCMC) for 1000 datasets generated as described in Section 4.3.1.

		Mean			Standard Deviation		
**Setting**	**Method**	$\omega^{1}$	$\omega^{2}$	ρ	$\omega^{1}$	$\omega^{2}$	ρ
ρ=0.1,β=(−1,1)	EM	8.55	10.41	0.107	9.30	11.67	0.139
	0	-	-	0.109	-	-	0.144
	SCOPE	9.13	12.25	0.105	8.47	11.00	0.144
	MCMC	18.21	24.99	0.094	16.62	21.89	0.127
ρ=0.5,β=(0,2)	EM	8.03	16.48	0.455	8.68	19.47	0.139
	0	-	-	0.498	-	-	0.138
	SCOPE	9.06	45.25	0.486	8.25	36.11	0.143
	MCMC	17.90	59.34	0.393	16.13	57.15	0.131

**Table 2 entropy-24-01849-t002:** Comparison of the algorithms with respect to the Cramer–von Mises distance between the estimated and the true first ($\omega^{1}$) and true second marginal distributions ($\omega^{2}$), as well as the correlation (ρ). Shown are the mean and standard deviation of the proposed EM algorithm (EM), the method based on known marginals (0), the Standard Copula Estimator (SCOPE), and the Markov chain–Monte Carlo approach (MCMC) for 1000 datasets generated as described in Section 4.2 with ρ=0.5 and β=(0,2) and varying sample sizes N=100,200,500,1000.

		Mean			Standard Deviation		
**N**	**Method**	$\omega^{1}$	$\omega^{2}$	ρ	$\omega^{1}$	$\omega^{2}$	ρ
N=100	EM	8.03	16.48	0.455	8.68	19.47	0.139
	0	-	-	0.498	-	-	0.138
	SCOPE	9.06	45.25	0.486	8.25	36.11	0.143
	MCMC	17.90	59.34	0.393	16.13	57.15	0.131
N=200	EM	4.91	8.53	0.469	5.46	8.88	0.098
	0	-	-	0.500	-	-	0.094
	SCOPE	4.76	37.38	0.493	4.18	25.35	0.096
	MCMC	9.27	42.91	0.370	8.01	36.23	0.089
N=500	EM	3.01	3.83	0.480	2.92	3.59	0.063
	0	-	-	0.499	-	-	0.060
	SCOPE	2.05	31.92	0.497	1.85	14.95	0.060
	MCMC	4.01	31.41	0.0360	3.49	20.51	0.051
N=1000	EM	2.25	2.74	0.486	1.92	2.40	0.047
	0	-	-	0.500	-	-	0.042
	SCOPE	1.08	30.60	0.499	0.93	11.13	0.043
	MCMC	1.99	28.13	0.365	1.84	14.49	0.037

**Table 3 entropy-24-01849-t003:** Comparison of the proposed EM algorithm with respect to the Cramer–von Mises distance between the estimated and the true first ($\omega^{1}$) and true second marginal distributions ($\omega^{2}$), as well as the correlation (ρ), for different numbers of mixtures *g* in Equation (Equation 15). Shown are the mean and standard deviation for g=5,15,30 and for 1000 datasets generated as described in Section 4.2 with ρ=0.5 and β=(0,2).

	Mean			Standard Deviation		
**# Mixtures**	$\omega^{1}$	$\omega^{2}$	ρ	$\omega^{1}$	$\omega^{2}$	ρ
g=5	13.82	16.38	0.469	14.17	19.69	0.145
g=15	8.03	16.48	0.455	8.68	19.47	0.139
g=30	7.17	18.73	0.454	7.48	20.98	0.140

**Table 4 entropy-24-01849-t004:** Comparison of the algorithms with respect to the run time in seconds. Shown are the mean and standard deviation of the proposed EM algorithm (EM) with the number of mixtures *g* set to 5,15,30, the Standard Copula Estimator (SCOPE), and the Markov chain–Monte Carlo approach (MCMC) for 1000 datasets generated as described in Section 4.2 with ρ=0.5 and β=(0,2).

	Run Time in Seconds	
**Method**	**Mean**	**Standard Deviation**
EM (g=5)	21.78	3.27
EM (g=15)	55.94	11.39
EM (g=30)	161.57	38.00
SCOPE	0.45	0.11
MCMC	12.98	0.87

**Table 5 entropy-24-01849-t005:** Comparison of the algorithms with respect to the Cramer–von Mises distance between the estimated and the true first marginal distribution ($\omega^{1}$), true second marginal distribution ($\omega^{2}$), and true third marginal distribution ($\omega^{3}$), as well as the correlation (ρ). Shown are the mean and standard deviation of the proposed EM algorithm (EM), the proposed EM algorithm with prior knowledge on the conditional independencies (KP, EM), the method based on known marginals (0), the Standard Copula Estimator (SCOPE), and the Markov chain–Monte Carlo approach (MCMC) for 1000 datasets generated as described in Section 4.4.

	Mean				Standard Deviation			
**Method**	$\omega^{1}$	$\omega^{2}$	$\omega^{3}$	$\left|\right|\hat{\Sigma }-\Sigma {\left|\right|}_{2}$	$\omega^{1}$	$\omega^{2}$	$\omega^{3}$	$\left|\right|\hat{\Sigma }-\Sigma {\left|\right|}_{2}$
EM	12.12	13.38	21.15	0.229	13.89	14.25	22.44	0.113
KP, EM	12.04	13.28	19.66	0.182	13.93	14.37	20.88	0.111
0	-	-	-	0.227	-	-	-	0.108
SCOPE	17.57	17.55	26.69	0.232	16.75	15.55	24.84	0.113
MCMC	36.85	35.70	80.22	0.263	32.82	33.24	78.57	0.140

**Table 6 entropy-24-01849-t006:** Comparison of the algorithms with respect to the Kullback–Leibler divergence ($D_{KL}$) between the true joint distribution (*F*) and the estimates. Shown are the mean and standard deviation of the proposed EM algorithm (EM) and the proposed EM algorithm with prior knowledge on the conditional independencies (KP, EM) for 1000 datasets generated as described in Section 4.4.

	Mean($D_{\mathrm{KL}}\left(F,\;\cdot \;\right)$)	Standard Deviation($D_{\mathrm{KL}}\left(F,\;\cdot \;\right)$)
EM	1.37	0.53
KP, EM	1.26	0.32

## Data Availability

The data generation procedures of the simulation studies and the proposed algorithm are availabe at https://github.com/mkrtl/misscop, accessed on 26 October 2022.

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
