# Peer review of "Estimating Gaussian Copulas with Missing Data with and without Expert Knowledge"

_entropy, 2022, doi:10.3390/e24121849_

Round 1

Reviewer 1 Report

The submitted manuscript proposes an EM algorithm for estimating the Gaussian copula model with missing data. In general, the paper is well structured. I found the theory part solid, but the simulation study was poorly designed and presented.
Major points:
1.    I found the wording “expert knowledge” in the title a bit ambiguous. First, one includes knowledge in the analysis by using known marginals or correlations into the analysis. I do not see why one should frame this as “expert knowledge”. What is an “expert”? Is there knowledge from “non-experts”?
2.    Section 4: A range of (larger) sample sizes is required to empirically demonstrate that the proposed estimator is consistent. Moreover, finite-sample bias could be of interest.
3.    Sec. 4: I think presenting simulation results only in figures is of no scientific value. Please present bias and RMSE (or some summary for the estimated marginals). Figures can be an add-on, but they should not replace numerical findings.
4.    Sec. 4: Also compare the MCMC method of Hoff (2007) and demonstrate that your proposed method performs superior.
5.    Sec. 4: Please empirically study the influence of the number of mixtures g. Study bias and RMSE as a number of g in the simulation. Maybe this issue is unrelated to the occurrence of missing data.
6.    The authors distinguished the cases of no knowledge or perfect knowledge. Likely, the latter situation is not really interesting. Typically, one would encode imperfect knowledge as prior distributions. The EM algorithm can be adapted to using prior distribution and maximum aposterior estimation instead of (constrained) maximum likelihood estimation could be utilized.
7.    Please include some discussion on how to obtain standard errors.
Minor points:
8.    26: language issue “In case missings are present …”
9.    27: language issue “On the other side …”
10.    59: Shouldn’t one say that the two steps “alternate”?
11.    Above Eq. (6): language “we can read off …”
12.    142: what is meant by “embedding of this intuition”?
13.    145: language issue: “in case of data MAR …”

Reviewer 2 Report

Contributions

The paper presents estimation of the parameters of Gaussian copulas with the Expectation Maximisation algorithm, working with missing data. Semi-parametric marginal distributions are included.  Expert knowledge is also included in the algorithm. the performance of the proposed algorithm is carefully evaluated and validated. 

The paper is well structured, well written and its novelty is clear.

Suggestions for improvement

Please discuss the computational complexity/ efficiency of the algorithm.

On Figures 5, 6 and 7, please indicate what is on the x and y axes, respectively. 

Round 2

Reviewer 1 Report

Thank you for the thorough revision of the paper. I do not have any further comments.